# What is the effect of secondary (high) schooling on subsequent medical school performance? A national, UK-based, cohort study

Lazaro M Mwandigha,[1] Paul A Tiffin,[1] Lewis W Paton,[1] Adetayo S Kasim,[2] Jan R Böhnke[1,3]

[1]Department of Health Sciences, University of York, Heslington, UK
[2]Wolfson Research Institute for Health and Wellbeing, Durham University, Stockton-on-Tees, UK
[3]Dundee Centre for Health and Related Research, School of Nursing and Health Sciences, University of Dundee, Dundee, UK

**Correspondence to**
Dr Paul A Tiffin;
paul.tiffin@york.ac.uk

## ABSTRACT

**Objectives** University academic achievement may be inversely related to the performance of the secondary (high) school an entrant attended. Indeed, some medical schools already offer 'grade discounts' to applicants from less well-performing schools. However, evidence to guide such policies is lacking. In this study, we analyse a national dataset in order to understand the relationship between the two main predictors of medical school admission in the UK (prior educational attainment (PEA) and performance on the United Kingdom Clinical Aptitude Test (UKCAT)) and subsequent undergraduate knowledge and skills-related outcomes analysed separately.

**Methods** The study was based on national selection data and linked medical school outcomes for knowledge and skills-based tests during the first five years of medical school. UKCAT scores and PEA grades were available for 2107 students enrolled at 18 medical schools. Models were developed to investigate the potential mediating role played by a student's previous secondary school's performance. Multilevel models were created to explore the influence of students' secondary schools on undergraduate achievement in medical school.

**Results** The ability of the UKCAT scores to predict undergraduate academic performance was significantly mediated by PEA in all five years of medical school. Undergraduate achievement was inversely related to secondary school-level performance. This effect waned over time and was less marked for skills, compared with undergraduate knowledge-based outcomes. Thus, the predictive value of secondary school grades was generally dependent on the secondary school in which they were obtained.

**Conclusions** The UKCAT scores added some value, above and beyond secondary school achievement, in predicting undergraduate performance, especially in the later years of study. Importantly, the findings suggest that the academic entry criteria should be relaxed for candidates applying from the least well performing secondary schools. In the UK, this would translate into a decrease of approximately one to two A-level grades.

## INTRODUCTION

Internationally, there is high competition for places to study medicine and the UK is no

exception. Along with the academic demands of medicine as a subject, this has driven medical schools to use secondary (high) school performance as a major determinant to offer a place or not. In general, relatively high obtained (or predicted) grades at senior school are required before a candidate is considered as a potential entrant to medical courses. This emphasis on *prior educational attainment* ('PEA'—the grades obtained at formal examinations during secondary education) has partly driven the over-representation of socioeconomically privileged individuals in medicine. For example, in North America, the majority of US medical school entrants are from relatively affluent backgrounds with around half coming from families in the top fifth for national income.[1] This issue is inevitably reflected in the educational backgrounds of students—it was recently highlighted that 80% of those studying medicine in the UK applied from only 20% of the country's secondary schools.[2] Most of the secondary schools that provide

### Strengths and limitations of this study

► Schools and university data were able to be linked permitting the first UK-based study that compared the academic performance of medical students drawn from poorly performing secondary schools against their counterparts from well-performing ones across all the 5 years of medical school.
► The sample was relatively large with a total of 2107 medical school students who matriculated in 2008 included in this study.
► The *skills* and *knowledge*-based undergraduate assessment outcomes are local, not nationally standardised measures.
► There were relatively high rates of missing data in the latter years of the study, especially in relation to undergraduate *skills*-based examinations.

medical students are selective schools, which are better resourced compared with the non-selective schools. Selective schools are also generally attended by students from more advantaged socio-economic backgrounds. Therefore, differences in performance between selective and non-selective schools reflect, to a high degree, differences in material deprivation rather than the intellectual ability of the students from those schools.[3]

It was partly with this in mind that 'aptitude' tests, mainly tapping into cognitive domains, were introduced into medical selection.[4] Such aptitude tests were first used to complement PEA in selection for undergraduate students in the USA in 1928 when the Medical College Admission Tests (MCAT) were developed to address high attrition rates in undergraduate medical school.[5][6] Since this time, the use of such tests for selection has spread to other parts of the world.[7–16] PEA has been demonstrated to have predictive validity for undergraduate medical school outcomes in Australia,[17] South Korea,[18] the UK,[19] Saudi Arabia,[20] India,[21] the Czech Republic[22] and New Zealand.[23] Aptitude tests such as the *MCAT* in the US[24] *Biomedical Admission Test (BMAT)* and *United Kingdom Clinical Aptitude Test (UKCAT)* in the UK,[3][12] *Undergraduate Medicine and Health Sciences Admission Test (UMAT)* in New Zealand,[25] *Hamburg Medical School Natural Science Test (HAM-Nat)* in Germany,[11] *Saudi National Aptitude Exam* in Saudi Arabia[20] and *the Health Professions Admission Test-Ireland (HPAT-Ireland)* in Ireland[26] have predictive validity for medical school outcomes. Indeed, some critics have highlighted that such aptitude tests may tap into similar constructs as traditional metrics of academic achievement such as high school grades. If this is the case, then such measures are unlikely to either facilitate widening access to medicine or add value within the selection process in general.

Some aptitude tests, such as the BMAT[9] and MCAT,[5] include sections evaluating semantic knowledge of the biomedical sciences. Performance on knowledge tests may predict undergraduate medical performance, at least in the early years, but are unlikely to add predictive value above and beyond traditional measures of academic attainment.[27] Other aptitude tests place more weight on evaluating fluid concepts of cognitive ability, such as the UKCAT.[10] In the case of the UKCAT some, although modest, ability to predict undergraduate performance, even after controlling for the effects of secondary school achievement, has been demonstrated.[28] However, it is currently unclear how the predictive abilities of the UKCAT are mediated by PEA and the extent to which this may vary across both the type of academic outcome and the 5-year period of undergraduate education in the UK. It has been further suggested that the UKCAT scores may be somewhat less sensitive to the type of secondary school attended, compared with the A-levels sat by students in England and Wales in their final year of schooling.[29] A-levels, usually in three subject areas, are generally undertaken in the last 2 years of secondary schooling and are roughly equivalent to Advance Placement courses

taken by some students in North America. Findings from an earlier, cross-sectional, study suggested that a strong use of the UKCAT scores during the admissions process may mitigate some of the disadvantage faced by certain under-represented groups applying to study medicine.[30] However, a subsequent study, using longitudinal data, did not report consistent effects over time in this regard.[31]

While PEA does predict academic outcomes in higher education, previous studies have observed an inverse relationship with the performance of the secondary (high) school attended, that is, students from more highly performing schools tend to get poorer degree awards, after controlling for PEA.[32] To date, the evidence relating to this potential effect in medical school has been inconsistent. One national study observed such an effect in the first year of medical undergraduate training for overall academic performance.[3] A separate, local study did not.[33] Certain medical courses, designed to widen access to medicine, already 'discounted' requirements for certain groups. For example, in Australia, a scheme to encourage recruitment to remote, underserved areas relaxes entry requirements for candidates from rural backgrounds.[34] In the USA, 'affirmative action' policies, although at times controversial and repeatedly legally challenged, have been implemented to encourage those from under-represented ethnic groups to enter medical school.[35] In the UK, a number of universities have started to offer reduced academic entry requirements for A-level (high school) grades to students from disadvantaged backgrounds who have attended poorly performing secondary schools.[36][37] Other medical schools are following suit.[38] However, evidence to support such admissions strategies is currently lacking. In the UK, individuals who wish to study at a UKCAT consortium medical school sit the test prior to making an application. The decision to make an offer, for those still at secondary school, is partly based on the predicted A-level (or equivalent) grades. This choice is commonly also informed by early achievement at the General Certificate of Secondary Education (GCSE) examinations, usually taken earlier in the applicant's school career. Therefore, any offers made would then be conditional on the specified scores obtained first at the UKCAT test before the end of secondary school and later grades being achieved at A-level at end of the secondary school education within each medical school selection cycle. Thus the present study had two aims:

1. To determine the extent to which the predictive powers of the UKCAT are mediated via PEA, for two separate domains (undergraduate *knowledge* and *skills*-based outcomes) over the period of undergraduate training. Since cognitive ability and educational attainment correlate, we attempt to achieve a more accurate assessment of the relative, and unique, contribution UKCAT scores make within the selection process.
2. To appraise the influence of the performance of the previous secondary school attended on an undergraduate's achievement in medical school. These results

will usefully inform policy on grade discounting for applicants applying from poorly performing schools.

For this study, we had an opportunity to link national data on the performance of secondary schools to cognitive ability (as evaluated via the UKCAT), PEA and outcomes at 18 UKCAT-consortium medical schools. Thus, there was also the possibility to better understand the interplay between secondary school-level performance, an individual's cognitive ability, their educational attainment (PEA) and how these related to subsequent undergraduate academic achievement. It was therefore hoped that a relatively sophisticated approach to modelling could help understand the role of secondary schooling in both selection (partly based on PEA and aptitude test scores) and later attainment at undergraduate level.

Our findings will inform selection policy in medical school and in particular provide guidance on the extent to which grades should be discounted for applicants from poorly performing secondary schools.

## METHODS
### Data availability and quality
UKCAT consortium medical schools are those medical schools that use the UKCAT for selection in the UK. For this study, data were available for 18 UKCAT consortium medical schools in England and Wales for candidates who were enrolled between 2007 and 2013. However, Department for Education data on the performance of English secondary schools were only linked to the 2008 entry cohort. For this reason, only data relating to these students were used in this study. It should be noted that an advantage of using the 2008 entry cohort was the relatively little missing data throughout the first four of the 5-year undergraduate period studied. In the 2007 UKCAT testing cycle, there were 26 UKCAT-consortium medical schools. Therefore, the data represented 69% of the 26 UKCAT-consortium medical schools. All medical school applicants who sat the UKCAT in 2007 and were selected to join one of the 18 UKCAT-consortium undergraduate medical schools in 2008 were included in this study. As with similar previous studies, non-standard medical courses (eg, 'widening participation', graduate entry and so on) were excluded.[28] Only the marks attained at first sittings of undergraduate examinations were retained for each student. Data relating to UKCAT scores and secondary school attainment were available for 2107 students who entered medical school in 2008 and had linked data relating to the performance of the secondary school they attended.

The secondary school examinations sat by the students were nationally standardised and included GCSE, Advanced Subsidiary (AS) Level and Advanced Level ('A-Level') examinations. The GCSE examinations are taken at around the age of 15–16 years. Those aspiring to eventually enter higher education usually take at least 10 subjects at GCSE level. At the time of the study, the AS levels were sometimes taken in the first year of sixth

form (equivalent to high school junior year) as preparation for or to supplement the full A-level examinations taken the subsequent year. For those planning to apply for medicine, three subjects at A-level are studied in the last 2 years of secondary schooling, almost always in the sciences. Candidates frequently take more than three A-levels though universities only count the highest three grades, that usually must be achieved at first sitting.

The completeness of the data relating to the outcomes of interest varied and the flow of the data in the study is depicted in figure 1.

The manner in which data related to undergraduate performance in the UKCAT consortium of universities have been collated and managed has been previously described.[28] However, to summarise, the main outcome variables used were the scores achieved at undergraduate *knowledge* and *skills*-based end of year outcomes. It was left to individual institutions to define how their assessments fell into each category. These assessment scores were provided by the universities in percentage forms (of maximum marks achievable) and then converted to standardised z-scores within each institution. Thus, the z-scores were created by subtracting the mean performance for that particular year and medical school cohort from an entrant's score and dividing it by the SD for their peers' scores. This created standardised scores with mean zero and an SD of one for each medical school group of students. This standardisation was carried out in order to minimise the impact of any variability across medical schools, in terms of the nature of the assessment.

The UKCAT, at the time of the study, consisted of four separate, timed, multiple choice subtests, namely *quantitative reasoning, decision analysis, verbal reasoning* and *abstract reasoning. Quantitative reasoning* assesses an applicant's ability to critically evaluate information presented in numerical form; *decision analysis* assesses the ability to make sound decisions and judgements using complex information; *verbal reasoning* assesses the ability to critically evaluate information that is presented in a written form and *abstract reasoning* assesses the use of convergent and divergent thinking to infer relationships from information. Each of the cognitive subtests has their raw score converted to a scale score that ranges from 300 to 900. Therefore, the total scale scores for all of the four subtests range from 1200 to 3600. The UKCAT subtests and their total scores were standardised as z-scores according to the scores for all candidates at the year of sitting. The reliability of the UKCAT subtests has previously been evaluated and reported.[39] For the purposes of this study, only the total UKCAT score (ie, the summed total of all four subtest scores) was used as a predictor. This is because it is the total score that is generally used in selection and represents a summary measure of all the four subtest scores. Full details of the descriptive statistics relating to total UKCAT scores are provided in section 1 of the online supplementary document.

In order to develop an overall, and precise, measure of PEA, we implemented a novel approach that extended

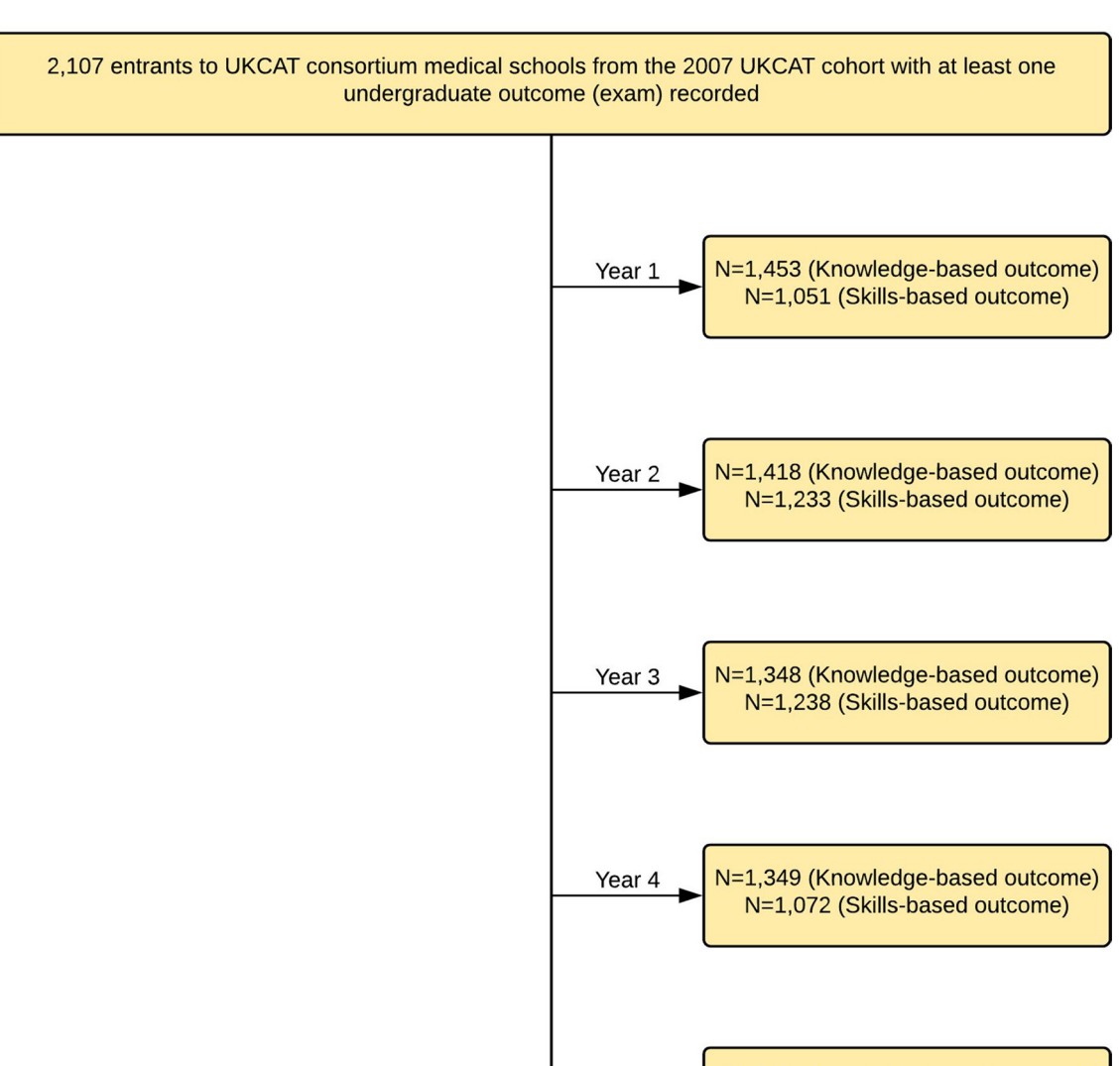

**Figure 1** Flowchart of data available for the outcomes for each of the five academic years of medical school training. UKCAT, United Kingdom Clinical Aptitude Test.

one previously used by McManus *et al.*[3 40] This involved conceptualising 'educational achievement' as a common factor ('latent trait'). Latent traits cannot be observed or measured directly, only by their effects on behaviour. In terms of attitudes, this could be observing certain responses to questionnaires, or in the case of ability, performance on examinations and other assessments. Thus, in this case, we treated all the commonly taken national examination grades (ie, GCSE, AS and A-levels) as 'indicators' (ie, observable markers) of an underlying ability (PEA). This approach allowed us to use information contained in all the commonly sat examinations during secondary school in England to estimate the overall underlying educational achievement of an entrant. Because the specific method we used easily accommodated missing 'indicators', it was irrelevant if only a minority of entrants had taken a specific exam (eg, history GCSE) and such grades could still be included when estimating PEA. The process resulted in a factor score estimate for each entrant which

was provided as a standardised z-score, where the mean was zero (average PEA for all applicants, with an SD of 1). Thus this measure of previous educational achievement provided more information on an individual than merely their 'best of three' A-level grades. Further details of the estimation of the PEA from the reported GCSEs, AS and A-level grades are provided in section 2 of the online supplementary document.

This estimate of PEA was used in the models addressing the first study aim (evaluating the mediating effects of previous educational attainment on the UKCAT's ability to predict undergraduate performance). However, 'discounting' policy focuses on the 'best of three' A-level grades required for entry, usually after a provisional offer has been made to an applicant. Therefore for the models addressing the study's second aim (role of secondary school-level performance on undergraduate outcomes), we banded entrants into categories according to A-level grades. Thus, the entrants were grouped into three bands

according to the highest three A-level grades achieved. Only 43 (2%) entrants were recorded as having the relatively low A-level grades 'BBB' and 'BBC'. Thus entrants were grouped into those with grades 'AAA', 'AAB' and 'ABB or lower'.

English secondary school-level performance data for 2008 were available from the Department for Education. Thus for this study, we defined secondary school-level performance as the average grades (converted to a numeric score) achieved for each student on roll at that educational establishment for that school year. Further details are available from the Department for Education for England website. In this sense, 'performance' is (narrowly) defined as the average educational attainment, in terms of formal examination grades achieved, for each student on roll, in that educational establishment.

### Data sharing statement
This study involved the analyses of anonymised secondary data of medical school entrants. Access to the data may be obtained from the UK Medical Education Database (http://www.ukmed.ac.uk) following approval of an application.

### Patient and public involvement
Patients, carers and members of the public were not involved in the design, conduct and analysis of this study.

### MODELLING APPROACHES
#### Modelling the relationship between UKCAT scores, PEA and undergraduate outcomes
Our first aim was to try and understand the extent to which the ability of the UKCAT scores to predict subsequent undergraduate medical school performance were explained by PEA. To answer this question, a mediation model was developed. The outcomes of interest (undergraduate *knowledge* and *skills*-based examination results) were local to each participating medical school. The variation in the assessment results across institutions was initially explored using a multilevel modelling approach, but no statistically significant clustering effects by university

were observed. For this reason, a simpler approach using a single-level mediation model was used for the analysis (figure 2). Further details of the single-level mediation model, the multilevel mediation model and rationale for choosing the single-level mediation model are described in section 3 of the online supplementary document.

#### Modelling the influence of secondary school performance on undergraduate outcomes
The second aim of this study was to evaluate the influence of the performance of an entrant's previous secondary school on subsequent undergraduate achievement. This involved estimating this secondary school-level effect while controlling for an entrant's A-level grades. A multilevel model was required to account for the variation in outcomes between universities.[41] Further details on the multilevel model can be found in section 4 of the online supplementary document. From the model, we could derive predictions about entrants' performances at medical school, for varying A-level grades and secondary school performance.

The statistical analyses were conducted using Mplus V.7.4, R and SAS softwares.[42–44] Lucidchart[45] was used to produce the figures and R software was used to generate the graphs of the model predictions.

### RESULTS
#### Descriptive statistics
The numbers of entrants with outcomes available in each category (type and year) are depicted in table 1. This was not a cohort study in the conventional sense (ie, entrants could leave and enter the study at any year based on a university deciding when to (not) report the academic outcome measures). Thus, table 1 also illustrates the missingness for only those entrants who had reported undergraduate *knowledge* and *skills*-based outcomes in the first year of undergraduate medical school. This is to provide a picture of attrition in the conventional sense (ie, how many participants at baseline remained at subsequent time-points).

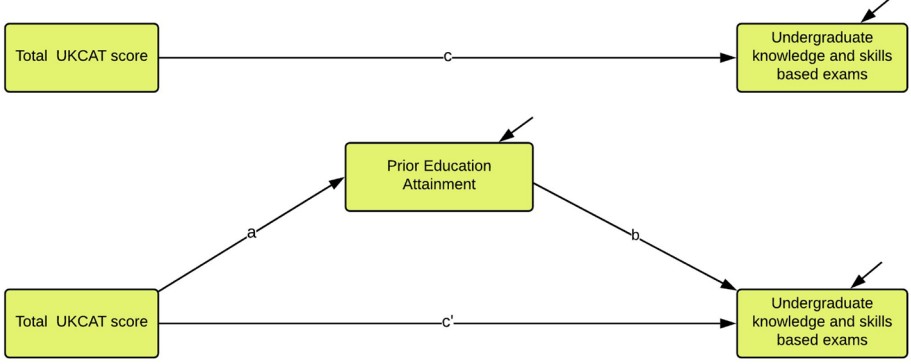

**Figure 2** Illustration of the conceptual model for the single level mediation effect of previous educational attainment on the association between total UKCAT scores and undergraduate medical school knowledge and skills-based examinations. UKCAT, United Kingdom Clinical Aptitude Test.

**Table 1** Study attrition rates due to missing data only for those students who had outcome measures reported in year one of medical school

| Academic Year | Undergraduate knowledge-based outcome | | | Undergraduate skills-based outcome | | |
|---|---|---|---|---|---|---|
| | Number of universities | Number of students | % Missing | Number of universities | Number of students | % Missing |
| 1 | 13 | 1453 | – | 9 | 1051 | – |
| 2 | 13 | 1404 | 3.37 | 9 | 1019 | 3.04 |
| 3 | 11 | 1041 | 28.36 | 7 | 729 | 30.64 |
| 4 | 7 | 711 | 51.07 | 5 | 668 | 36.44 |
| 5 | 4 | 439 | 69.79 | 2 | 260 | 75.26 |

Section 5 in the online supplementary document provides a detailed summary of the missing data patterns for the outcomes. Of the 2107 undergraduate medical school entrants, 1855 had their secondary school-level performance available. The distribution of secondary school-level performance and UKCAT scores achieved by the entrants are depicted in table 2.

Table 3 shows the distribution of A-level grades for the medical school entrants. Note that the majority of the entrants had achieved either AAA or AAB grades at A-level.

### The prediction of medical school outcomes from UKCAT performance

Figure 3 summarises the results from the models investigating the potential mediating effects of PEA on the relationship between UKCAT scores and undergraduate examination outcomes. The proportion of the predictive power of the UKCAT scores explained by PEA shown for both undergraduate *knowledge* and *skills*-based medical school outcomes are computed as a quotient of the indirect effect of UKCAT performance through PEA divided by the total effect of the UKCAT performance.

Overall, PEA explains approximately over 43% (dotted black line in the figure 3) of the statistically significant predictive power of the UKCAT for both undergraduate *knowledge* and *skills*-based examinations only in the preclinical years (one and two) of medical school training. For the clinical years (three to five), PEA explains approximately less than 43% of the predictive power of the UKCAT for both undergraduate *knowledge* and *skills*-based examination outcomes. This proportion remains statistically significant but declines somewhat with every subsequent year of training.

### The effect of secondary school-level performance on subsequent medical school performance

Both secondary school-level performance and PEA were statistically significantly related to the undergraduate outcomes. No statistically significant interaction was observed between the two variables. Overall, compared with entrants from secondary schools with a high average student performance, those from schools with lower average attainment tended to have better subsequent scores in both undergraduate *knowledge* and *skills*-based examinations. Lower levels of secondary school level performance corresponded with higher standardised undergraduate medical school performance as may be observed in figures 4 and 5.

We intended to make our results relevant to UK medical selectors. Specifically we wished to estimate the level of 'discounting' that should be offered to applicants from disadvantaged educational backgrounds. Thus the results of our models addressing the second study aim are depicted in figures 4 and 5. We show the actual and predicted (fitted) values from the models in the figures. Average secondary school performance (mean enrolled student attainment for all secondary schools in England) is shown on the horizontal axis and predicted medical school performance (as a standardised z-score) on the vertical axis.

Figure 4 depicts the values in relation to *knowledge*-based examinations, according to secondary school-level performance. Similarly, figure 5 shows the values for undergraduate *skills*-based outcomes. Superimposed on these plotted values are the estimates (with associated 95% confidence bands) for entrants depending on their A-level grades at admission to university. These

**Table 2** Descriptive statistics of the UKCAT total score and average point entry for the 2107 entrants from the 987 schools

**Year of UKCAT sitting=2007**

| | Sample size | Mean | SD | Minimum | Maximum |
|---|---|---|---|---|---|
| Average Secondary School-level performance | 1855 | 225.18 | 20.09 | 145 | 267.5 |
| UKCAT total score | 2107 | 2544.47 | 188.92 | 1950 | 3190 |

UKCAT, United Kingdom Clinical Aptitude Test.

**Table 3** A-level grades for the entrants in the study sample

| Grade | N (%) |
|---|---|
| Missing | 36 (1.71) |
| AAA | 1463 (69.44) |
| AAB | 436 (20.69) |
| ABB | 129 (6.12) |
| BBB | 29 (1.38) |
| BBC | 14 (0.66) |

represent the entrants within the three bands of A-level attainment ('AAA', 'AAB' and 'ABB or lower'). For purpose of demonstration, the horizontal black dotted lines indicate the equivalent level of performance between those entrants from secondary schools at the lower decile of performance and those at the upper decile.

There are a number of notable trends observed in these graphs. First, students with higher A-level grades outperform those with lower educational achievement. However, this gap narrows when predicting undergraduate *skills*, rather than undergraduate *knowledge*-based outcomes in medical school. The difference also reduces in magnitude as undergraduate education progresses through the years. Indeed for undergraduate *skills*-based outcomes and for many of the later years, the CIs for the groups' estimates generally overlap. This indicates no statistically significant intergroup differences between those with 'AAB' and 'ABB or lower grades' at the 95% CI.

The second most striking feature, and the focus of this study, is that students from less highly performing secondary schools generally outperform those from more highly performing educational institutions for any given A-level grade banding. That is, controlling for the effects of A-level attainment, on average, those from the more poorly performing schools tend to achieve better

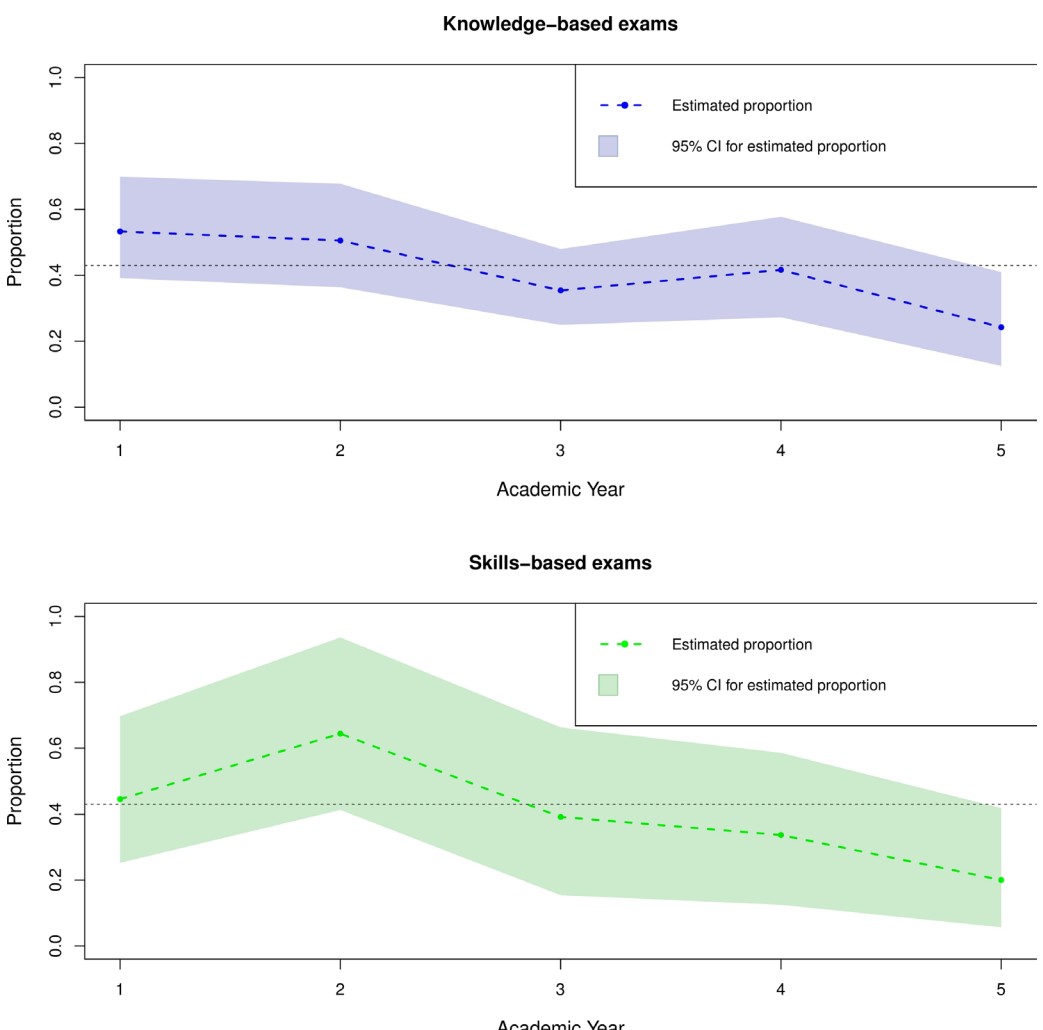

**Figure 3** Proportion of the predictive power of the UKCAT scores for undergraduate knowledge and skills-based examination outcomes explained by PEA in medical school. The proportion is computed as a quotient of the indirect effect of the UKCAT performance through PEA divided by the total effect of UKCAT performance. The black dotted line denotes the threshold at 43% selected so as to contrast the trend between the 'preclinical' (first two) years and the 'clinical' years (three to five) of medical school undergraduate training. PEA, prior educational attainment; UKCAT, United Kingdom Clinical Aptitude Test.

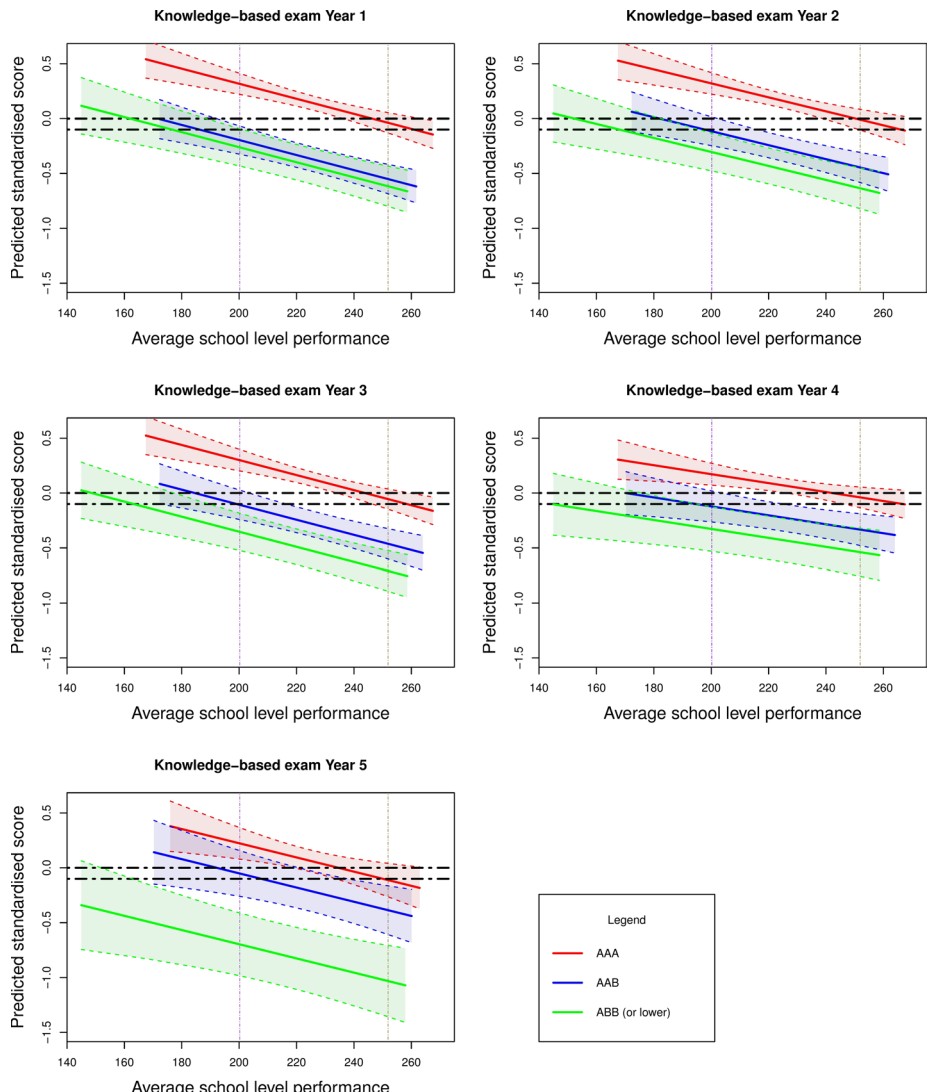

**Figure 4** Effect of average school level performance by reported grades on undergraduate medical school knowledge-based examinations (as a standardised z-score) for all secondary schools in England in 2008. The second decile (average school level performance of 200.2) and eighth decile (average school level performance of 251.9) are denoted by the purple and brown vertical lines, respectively. The horizontal black dotted lines are arbitrary points chosen to indicate the equivalent level of performance between those entrants from secondary schools at the lower decile of performance and those at the upper decile of performance.

undergraduate examination results than those from the schools with higher levels of student attainment. The vertical purple and brown dotted lines highlight this feature. They show that those with lower A-level grades (eg, AAB or ABB) from the lowest performing secondary schools tend to have equivalent undergraduate performance to those entrants from the highest performing educational establishments with top grades (ie, AAA). It is also notable that this 'secondary school gradient' is generally steepest for undergraduate *knowledge*-based outcomes in the early years of undergraduate study. Thus, the effects of secondary school environment, as with individual previous educational attainment, tend to be less marked for procedural (undergraduate *skills*-based) learning and with advancing time in university study.

## DISCUSSION

The findings from previous studies suggested some modest added value of the UKCAT scores to predict undergraduate performance, over and above that provided by conventional measures of academic achievement.[3 28] Further, the ability of UKCAT scores to predict certain aspects of undergraduate performance was found to be largely independent of PEA. This was less true for both undergraduate *knowledge* and *skills*-based examinations, taken early on in the preclinical years of medical school, where a significant portion of the UKCAT's predictive ability is mediated via previous educational performance.

Our findings on the role of secondary school quality in determining subsequent undergraduate performance are in line with the findings from a previous national study using data from the same cohort as well

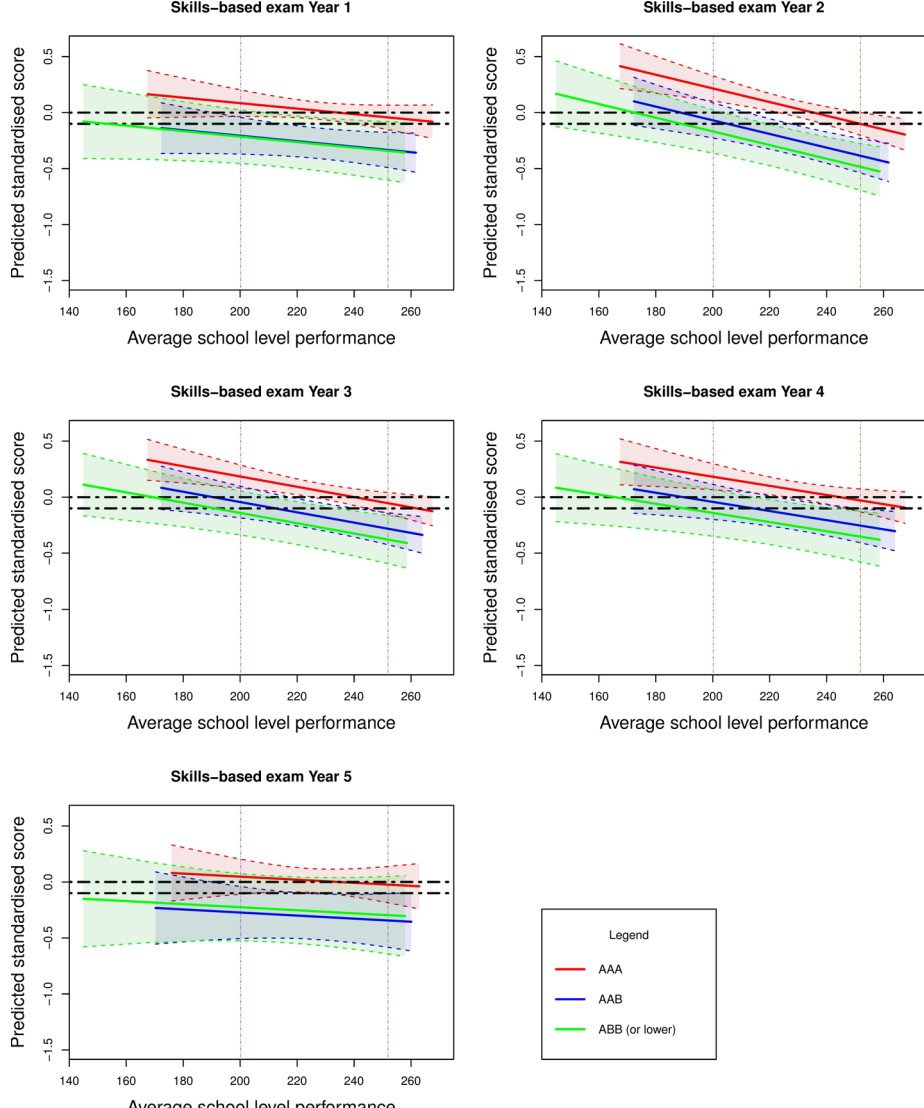

**Figure 5** Effect of average school level performance by reported grades on undergraduate medical school skills-based examinations (as a standardised z-score) for all secondary schools in England in 2008. The second decile (average school level performance of 200.2) and eighth decile (average school level performance of 251.9) are denoted by the purple and brown vertical lines, respectively. The horizontal black dotted lines are arbitrary points chosen to indicate the equivalent level of performance between those entrants from secondary schools at the lower decile of performance and those at the upper decile of performance.

as more general analysis of data from higher education in England.[3 32] However, we were able to demonstrate persistence (though attenuation) of these effects over the 5 years of medical school. It is also in keeping with recently published findings that showed that medical students from state-funded (mainly non-selective) secondary schools tended to academically outperform those from privately funded schools, once at university.[46] Our findings were also consistent with those from an Australian study. This reported that entrants from rural backgrounds tended to have lower educational achievement, both at entry and in the early, preclinical years of study. However, there were no significant intergroup differences in performance observed in the latter, clinical years of undergraduate training. However, some caution must be exercised in interpreting these findings

as the study was single site with a relatively small number (n=856) of participating students.[34] The present findings were in contrast to those of a local study, which focused on the fourth year of medical school, when the effects of secondary schooling are likely to have been less marked.[33] The relatively low numbers of students (n=574) involved in this latter study may have led to a deficiency in study power and thus an inability to demonstrate these effects. Also, by using a more sophisticated approach to statistical modelling, we were able to delineate the direct and indirect (mediational) effects of cognitive ability (as assessed via the UKCAT) in determining undergraduate medical academic performance. This highlighted the shifting relative roles that conventional academic achievement versus cognitive ability play as undergraduate training progresses. We were also able to separate, at least crudely,

undergraduate outcomes in this study relating to 'knowl-edge' and 'skills' (see also limitations, below). As expected, traditional academic attainment (in the form of PEA) was more predictive and mediated a greater proportion of the UKCAT effects for earlier examination performance. We also observed a narrowing of the effects of secondary education achievement as medical school progressed. This might be expected as the time since leaving secondary schooling elapses, it becomes less relevant to current academic performance. However, this narrowing gap may be due to a positive influence of the university educational environment, which may render prior disparities in educational achievement between students less influential. Alternatively, the shrinking disparity may be, at least in part, due to the students becoming more homogenous over time. Some, less well-performing or motivated students, will leave the courses in earlier years. Nevertheless, in the UK, as elsewhere, such medical school attrition rates (for all reasons) are very small, ranging from approximately 0.25% for the first year to 0.1% for the final year, for standard entry courses.[47] Therefore, this effect will have been only slight. In addition, as medical school progresses, there is an increasing emphasis on procedural (undergraduate skills-based) learning. Thus, the academic abilities required to highly achieve at written school examinations are likely to become less relevant to performance.

Our findings also build on previous research[3] and we were able to demonstrate the value, to some extent of 'contextualising' secondary school achievement across the medical undergraduate years. That is, to some extent, the grades obtained by a student at secondary school must be put in the context of the educational establishment in which they were obtained. A reduction of one to two A-level grades may not appear to be a large adjustment. However, this must be understood in the light of the highly homogenous nature of both medical school applicants and entrants where high proportions obtain the maximum achievable grades. Thus, even one grade difference could represent an SD from the mean in a pool of high achieving medical school entrants. Internationally, selectors must understand their equivalent effects, not just for school-type attended, but a range of contextual factors that may be pertinent to their culture. Similarly, they must translate such effects into discounted offers where appropriate, in the metric of their own educational systems.

The main strength of this study is that there were a relatively large number of entrants studied from a range of UK medical schools involved. This provided sufficient power to enable the elicitation of relatively subtle effects and suggests the findings are generalisable to England and Wales. Moreover, the secondary school examinations sat by this cohort were nationally standardised, with only a minority of the credits awarded for course work. Thus, any local or regional variation in standards can be assumed to be trivial. Nevertheless, a number of limitations must be borne in mind when interpreting the findings. In terms

of the outcome measures, the categorisation of undergraduate examinations into skills and knowledge was not operationalised and therefore rely on the participating medical schools to categorise the evaluations. Thus, their definition may vary across medical schools. While some of this variation was handled by the use of multi-level modelling, a more robust definition of undergraduate 'skills'-based assessments may have been helpful in predicting clinically orientated performance, which may have been a more faithful proxy for later medical practice. In this regard, a methodology has been proposed to achieve this through the 'nationalisation' of 'local' measures of undergraduate medical school performance for fair comparisons of graduating medical doctors.[48] It is also acknowledged that it is generally the case that undergraduate skills-based examinations are less reliable than knowledge-based tests.[49] It is thus possible that this likely disparity in reliability may explain the difference in the magnitude of observed relationships associated between the predictors and the two undergraduate medical school outcomes. Thus, lower reliability in the measurement of an outcome would have an attenuating effect on observed strength of the relationship.[19] In addition, scores from the most recently taken UKCAT scores were used. Scores from the first sitting of the test may have been a better metric of underlying cognitive ability (being less prone to practice effects). However, some early sittings may have been used as 'practice runs' by medical school applicants. In addition, the most recent UKCAT test results are those used by selectors, thus the ones most relevant to selection policy.

The number of participating universities in the study varied from year to year with higher levels of missing data for undergraduate skills-based assessments (compared with knowledge) and for the latter years of study. This was a result of medical schools deciding not to return outcome examination data for that year rather than students exiting the study or dropping out from medical school. Therefore, such missing data are likely to be 'missing completely at random' (MCAR- that is missing purely by chance) or potentially 'missing at random' (MAR- that is missing values are related to those that can be observed). This was dealt with by modelling the data using a likelihood approach and conducting sensitivity analysis to determine the effect of missingness through Multiple Imputation. Both likelihood modelling approach and Multiple Imputation are valid data handling methods, assuming the missing data are either MCAR or MAR.[50 51] The results from imputed versus non-imputed datasets can be compared as a form of sensitivity analysis (see section 6 of the online supplementary document). These highlight that the results did not vary significantly between imputed versus non-imputed datasets. Therefore, missing data did not adversely impact the results and conclusion of the study.

The quality of secondary schools previously attended by undergraduate medical school entrants varies widely across the UK. However, it is known that 80% of UK medical

students come from 20% of secondary schools[2] and tend to come from economically advantaged backgrounds.[52] Thus, students from selective, academically high-performing schools are grossly over-represented at medical school. Indeed, a selection process substantially based on predicted or actual A-level performance will greatly advantage applicants from such educational institutions. Paradoxically such students, once admitted, may relatively underperform in medical school, compared with their contemporaries from less well performing schools, which tend to be state funded and non-selective in nature. Already some UK medical schools are offering 'discounted' A-level offers to applicants from schools that have students with lower levels of academic attainment.[53–55] Our results suggest that such medical schools may have been (although serendipitously) implementing such policies broadly in line with our present findings. That is to say, entrants from the most poorly performing schools have achieved A-level outcomes 'worth' one to two grades more than those from the top performing schools, in terms of their ability to predict undergraduate achievement. As can be seen from figures 4 and 5, the definition of 'low' and 'high' performing secondary school is somewhat subjective. In addition, the suggested 'discounting' would vary according to the outcome of interest. There are also practical challenges to implementing such policies. Not all applicants to medical school will have attended schools which can supply comparable data on their institutional performance. At present, even comparison across the three nations making up the UK would be very difficult. One simple way of 'equating' across countries might be to report an applicant's rank within their school. However, further evaluation would have to be performed to assess whether such a relatively crude approach was an effective way of contextualising educational achievement. There is also the possibility of 'gaming' with economically advantaged families strategically placing a student in a less well-performing educational institution for the final year of schooling.

Any moves to widen access to medicine may prove controversial, as advantaging certain candidates necessarily means disadvantaging others. Thus, such policies must be based on defensible evidence, such as the kind we believe is offered by this study. Moreover, given the very low absolute numbers of applicants and entrants to medical schools from disadvantaged socioeconomic backgrounds, only a radical rethinking of 'widening access' is likely to result in substantial changes to the demographics of the medical workforce.

To conclude, we found that the predictive ability of the UKCAT can be explained to some degree by PEA, although this is more pronounced in the early preclinical years of undergraduate school. Significant effects of secondary school-level performance exist which suggest the issue of whether offers of a place to study should be discounted for students from more poorly performing schools. This highlights an urgent need to 'contextualise' secondary school performance in applicants rather than selectors taking grades at face value.

**Acknowledgements** The authors thank Rachel Greatrix at UKCAT for assistance with facilitating access to the data used in this study.

**Contributors** All authors made substantial contribution to this study. LMM conducted the statistical analyses and contributed to the writing of the article. PAT and ASK led the conception, design, supervision of the statistical analyses, interpretation of the results and contributed to the writing of the article. JRB was involved with supervision of the statistical analysis and handling of missing data, interpretation of the results, revising and writing of the manuscript. LWP was involved in the interpretation of the results, revising, writing and critical appraisal of the manuscript. All authors have approved the final version of the article submitted.

**Funding** LMM is supported in his PhD, of which this study is a component, via funding from the UKCAT Board. In addition, Hull York Medical School financially contributed to the student fees for LMM. PAT is currently supported in his research by a National Institute for Healthcare Research (NIHR) Career Development Fellowship. PAT is also lead for the DREAMS Network, an international collaboration on selection into the professions, of which LMM, JRB and LWP are also members, which is funded by a Worldwide University Network (WUN) Research Development Fund award. This paper presents independent research part-funded by the National Institute for Health Research (NIHR).

**Disclaimer** The views expressed are those of the authors and not necessarily those of the NHS, the NIHR or the Department of Health.

**Competing interests** LMM is supported in his PhD project via funding from the UKCAT Board and has received travel expenses incurred for attending a UKCAT Research Group meeting. PAT has previously received research funding from the Economic and Social Research Council (ESRC), the Engineering and Physical Sciences Council (EPSRC), the Department of Health for England, the UKCAT Board and the General Medical Council (GMC). In addition, PAT has previously performed consultancy work on behalf of his employing University for the UKCAT Board and Work Psychology Group and has received travel and subsistence expenses for attendance at the UKCAT Research Group.

**Patient consent** Not required.

**Ethics approval** No human subjects were tested for this study, therefore no ethical approval was necessary. The anonymised raw data used in these analyses were made available by the UK Medical Education Database (www.ukmed.ac.uk) following approval of an application. From the data, no piece of information can be used to identify a secondary school, medical school or individual. As such, the identity of the participants is fully protected. All participants consented to the collection of the data for research.

**Provenance and peer review** Not commissioned; externally peer reviewed.

**Data sharing statement** This study involved the analyses of anonymised secondary data of medical school entrants. Access to the data may be obtained from the UK Medical Education Database (www.ukmed.ac.uk) following approval of an application.

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
