## [Reviewer comments · BMJ Open]

ARTICLE DETAILS

TITLE (PROVISIONAL)	What is the effect of secondary (high) schooling on subsequent medical school performance? A national, UK-based, cohort study
AUTHORS	Mwandigha, Lazaro; Tiffin, Paul; Paton, Lewis; Kasim, Adetayo; Boehnke, Jan Rasmus

VERSION 1 – REVIEW

REVIEWER	John Norcini FAIMER, USA
REVIEW RETURNED	03-Nov-2017

GENERAL COMMENTS	The aim of this study was to understand the relationship of educational achievement and scores on UKCAT with subsequent medical school performance. Using linked datasets, information based on more than 2000 students was analyzed. The results indicated that the UKCAT scores predicted knowledge-based medical school performance mediated by PEA; the effect was less pronounced for skills-based performance. For both outcomes, the effect diminished over time. The authors conclude that entry criteria should be relaxed for students from less well performing secondary schools. These are important questions and an excellent linked set of data with which to answer them. The introduction is comprehensive, the data are analyzed in a very sophisticated fashion, and the conclusions have serious implications. As with any observational study, however, there are a variety of alternatives to consider. First, it would not be unusual for a skills-based final exam to be less reliable than a knowledge-based examination. Lower reliability has the effect of attenuating the magnitude of relationships and could explain the difference between skills- and knowledge-based assessments. At least, this must be acknowledged. Second, it would be helpful to get some sense of the differences among the medical schools in UKCAT and PEA. I would expect that some were more selective than others. Likewise, I suspect that the schools varied in the number of students they contributed to the analysis. How does this influence the results? Third, the outcome measures were standardized within school. This removes any variability associated with differences among them, although these differences are present in the predictor variables. How does this influence the results, particularly for those from less well performing secondary schools?
--

	Fourth, as the authors note this is not a typical cohort study in that students from the 2008 group enter and exit medical school at different times. It would be useful to provide more descriptive information (perhaps in the appendix) about which students (by PEA, UKCAT) enter sooner rather than later (do less able students enter later?). Why is there student attrition? Is attrition related to PEA or UKCAT scores?
--	---

REVIEWER	Associate Professor Deborah O'Mara Sydney Medical School University of Sydney, Australia
REVIEW RETURNED	05-Nov-2017

GENERAL COMMENTS	This is a substantive, thorough and statistical rigorous research study which contributes to the existing literature on the impact of educational attainment on medical school academic and clinical performance. As such it adds to the knowledge base of selection for medical education research and practice. The high attrition rate is disappointing for year 5, and I wondered why the authors did not restrict the analysis to Years 1-3/4. I was unable to find an explanation for the high attrition, was it the unavailability of medical school assessments or did some of the 2008 cohort start medical school early? Please make this clearer. Clinical skills assessments converted to a percentage score are invariably highly negatively skewed in medical education and converting to a zscore does not turn this into a normal distribution. Please provide some graphical illustration of the actual distributions of the scores you use. The technical appendix is very lengthy and could benefit from an edit to make it more parsimonious. I expected more information about the differences in medical schools performance data and their distributions and less emphasis on statistical formulae. Is it possible that Figure 4 is incorrect? – it seems identical to Figure 3 and all other figures for “theory” (assume knowledge) and “skills” are different. The main body of the text could be more parsimonious as well. For example if you had no intention of using the sub-score of the UKCAT, why explain them in such detail? I would suggest the following minor editorials changes prior to publication:  • Typo in the first sentence of the abstract if I am not mistaken: Should the third last word be at rather than an? • The first paragraph of “strengths of the study” needs as edit as well. • Page 13, line 50 the second this should probably be the • Page 18, line 34 should read tend to • Be consistent in the use of knowledge and skills – theory and skills is used in some places and for some graph
--

REVIEWER	Samuel Tomczyk University of Greifswald Institute of Psychology Department of Health and Prevention
-----------------	--

	Robert-Blum-Straße 13 17487 Greifswald Germany
REVIEW RETURNED	22-Nov-2017

GENERAL COMMENTS	The authors investigate the associations between the United Kingdom Clinical Aptitude Test (UKCAT), prior educational attainment (PEA), secondary school performance at the school level, and results of undergraduate medical exams (knowledge and skills). Their findings highlight the role of PEA in the prediction of academic success via UKCAT scores and they point to differences between individual performances based on school-level performance. I think that this manuscript addresses a very important issue and I would consider it a substantial contribution to the discussion about university/medical school entrance policy. However, while I appreciate the sophisticated methodological approach and the extensive and intelligible supplementary material, I do have some questions and concerns, particularly regarding the introduction and the methods. Personally, I think that the manuscript can benefit from some restructuring and rephrasing. Abstract  1. I do not think that the abstract is entirely correct. Exploring “the mediating role played by the performance of a candidate’s secondary school” (p. 2) is listed as a main objective. However, the authors did not investigate school-level performance as a mediator, but rather a predictor of undergraduate outcomes (knowledge and skills). I would suggest rephrasing this section, for instance: “Moreover, we explore the impact of school-level performance on undergraduate outcomes (knowledge and skills) to inform selection policy”. 2. In addition, the authors should state that they investigate knowledge and skills as two separate outcomes in their study. Currently, this distinction appears in the Results for the first time. 3. Under Methods, the authors could state that they used data from the 2008 medical school entry cohort for all purposes. I think that it is confusing to explicitly link “[m]edical school outcomes” (p. 2), but neither UKCAT scores nor PEA to the year 2008. Moreover, I do not think that there is a sufficient distinction between “secondary school exam grades” and “school-level performance data” (p. 2). I was puzzled, since I expected a mediation including medical school outcomes, UKCAT, and PEA, but not necessarily school-level performance data. In accordance with “Objectives”, I would recommend rephrasing the section, for example, “UKCAT scores and Prior Educational Attainment (PEA) were available for 2,107 students and were linked to medical school outcomes for each of the five years of medical school. PEA and school-level performance were based on school exam grades.” Thus, it becomes clear that both, PEA and school-level performance are based on grades. It takes some time and space to explain the difference in the operationalization of these two, but I think that the main text is the place to do so rather than the abstract. 4. Similarly, I would state something like “knowledge-based undergraduate performance” rather than “exam performance” to maintain clarity in that “school exam grades” are not the same as (undergraduate) exams.
--

5. Moreover, the authors do not mention the mediation of UKCAT scores by PEA for skills-based exam performance. This should be mentioned, because it refers to the main research question.

Introduction

6. I am not sure why high grades are supposed to be connected to socio-economic status or even the type of school. Theoretically, students at underperforming schools, i.e. schools with a school-level performance below average, can achieve high grades and attend medical school. It seems clear to me though, why 80% of medical students stem from 20% of the country's schools, if these schools belong to the high performance schools in the country, and most of their students achieve high grades. In my opinion, a major flaw in a lot of these estimations and policies is the misrepresentation of school-level performance as a sociocultural factor, which is one of the strengths of this study. Because a discounting procedure for individual applicants that is based on school-level performance, does not accurately consider the differences between schools and clustering effects of schools. Thus, I would suggest building a stronger argument for the differentiation of individuals and schools in "widening access" to medical education. Because apart from mentioning in the introduction, the authors do not assess socio-economic differences and consequently, do not include it into their model.

7. Another point refers to the similarities of PEA and UKCAT. I am not sure if the authors wanted to stress a difference by stating that "PEA has been demonstrated to have predictive validity for undergraduate medical school outcomes" (p. 4) and "Aptitude tests [...] have predictive validity for medical school outcomes" (p. 5). Maybe I have missed the point here, but as I see it, the authors point to potential similarities between both tests without additional value. To make this point more convincing, the authors could reference national data to compare entrants or applicants between schools with and without aptitude tests. If aptitude tests were invented to address the gap between socioeconomically advantaged and disadvantaged students or schools, then the gap between entrants for low-performance schools and high-performance schools should shrink after aptitude tests have been introduced. Unless there are other factors, such as sociocultural influences that are more important when deciding to apply for medical school.

8. Additionally, the lack of content validity could be discussed separately. A first step could be to provide measures of correlation between both, UKCAT and PEA or an estimate of their predictive value regarding undergraduate achievement.

9. Unfortunately, I am not very familiar with the educational system in the UK. Therefore, I would appreciate if the authors simply stated the order of these exams and tests at one point. They could simply outline that UKCAT are taken before final exams (secondary school exams) and before medical school, i.e. in the summer of the year before entry, I believe. In addition, they could make clear that they estimated PEA on standard exams like GCSE and A-levels in contrast to general school performance (predicted grades). In their present analysis, if I am correct, the authors investigate PEA, based on final secondary-school exams, as a mediator of UKCAT scores,

which were assessed at an earlier time, on medical school performance (undergraduate performance at the end of the year). I think that it would be much easier for an international audience to comprehend the mediation models and the interplay of these variables, if the authors provided a clear description and a flow.

10. I have a small issue with the wording in the main text. The authors discern school-level performance (ordinal grades like AAA, ABB) and PEA (latent variable) as indicators of school performance on a school-level and an individual level. However, this could be solved by introducing a fixed set of words from the beginning. If they referred to school-level performance and individual performance or PEA, for example, it would be much easier not to confuse these two levels when reading the methods, and results. Similarly, the authors could decide to use “medical school” or “undergraduate” outcomes, such as “undergraduate knowledge” and “undergraduate skills”. However, I think that they should not switch between these phrases within the text. In my eyes, the sophisticated analysis is much easier to comprehend if presented consistently.

11. I do not see why the influence of school-level performance on undergraduate’s achievement should necessarily inform policy on grade discounting. In the current sample, the authors investigate students from poorly performing schools that have already taken UKCAT and “survived” medical school for at least a year. This could mean that they have a different outlook on medical school compared to their peers or that they feel the need to prove themselves among socio-economically advantaged peers, which would explain their high scores and test results. Thus, it is hard to compare them to the average at their secondary schools. If these students already represent “the best” at their school, it could be possible that grade discounting for their secondary school does not lead to the expected consequences. Instead, I think that it could be interesting to compare UKCAT scores of students from schools with different performance levels and estimate the predictive value of these scores for academic success. For analytical purposes, the authors could suggest a comparison of schools, e.g. between two or three groups (+1 SD and -1SD of school-level performance), and report coefficients for these groups. Thus, they could also report the number of students at AAA and AAB levels in each group, and so on to further elucidate these differences.

12. I would like to know how many UKCAT-consortium medical schools there are. It should be possible to report how many schools and students are represented by the 18 schools and 2,107 students included.

Methods

13. Although the authors did not recruit the sample themselves, they should provide some information regarding data privacy standards, and ethical considerations. The UKCAT consortium may have addressed these points in the past, but it is important to read and comprehend the researchers’ point of view and possible concerns.

14. The authors could state the exact value for the “relatively low attrition rate” (p. 8) in the text.

15. I am curious: Was it possible to assess sociodemographic information like race/ethnicity, gender or age? If not, this should be discussed as a limitation.

16. One of my main concerns refers to the centering process used in this study. I appreciate that the authors provide information on the preparation of their data set, but I am not sure I understand every step. For example, medical school outcomes were assessed as z-scores for each school (group-mean centering), so that each students z core can be compared to its peers, but not across schools. Thus, differences in “high performance” medical schools or cohorts at some schools may lead to similar z scores compared to other schools with lower grades due to the reduced variance of the subsample. Unfortunately, the authors do not provide variance estimates for each medical school so it remains unclear. However, they state intraclass correlations (ICC) of outcomes in their appendix. First, I think that this information is important and belongs in the Methods’ main text. Second, I am not sure if the ICC are based on percentage scores or z scores.

17. In contrast, the UKCAT scores were standardized “for all candidates at the year of sitting” (p. 10) as were PEA factor scores “for all applicants” (p. 11) (grand-mean centering). However, this means that UKCAT and PEA are compared between all of the students in the sample regardless of their secondary school and their medical school. Thus, it is not possible to deduce whether a student at a poorly performing school belonged to the best in his or her secondary school (which is problematic regarding the second research question), and whether there are differences in PEA and UKCAT scores between medical schools that could also correspond with school-level performance. To investigate these differences, the authors could also provide ICC estimates for raw scores. Finally, school-level performance is stated as a school-level grade average without centering. Thus, differences in school-level performances reflect mean differences for a student, irrespective of the variance of performance at each institution. Thus, school-level performance cannot be easily compared to PEA, UKCAT scores or outcomes (all of which are centered in one way or another). I wonder if the results differed if grand-mean centering was applied to school-level performance, so that the mean value of 225.18 points referred to “0” or average performance, and higher and lower scores represented “higher” and “lower” performance across schools.

Results

18. This points to the multilevel models, in general. In their technical appendix, the authors explain that neither effect varied significantly between medical schools. However, they do not report their findings for educational achievement, school-level performance, and so on in the main text. While I appreciate the detailed explanation in the supplementary material, I would recommend including more information in the main text to guide the reader through each decisional step.

19. I am confused by the choice of words. First, there are “no statistically significant clustering effects by university” (p. 13), but then there is “variation in outcomes between universities” (p. 13) that needs to be accounted for. I assumed that in both cases, universities represented the higher level, and medical school outcomes represented individual “undergraduate achievement” (p.13). If the

authors refer to schools instead of universities or to school-performance instead of undergraduate achievement, they should say so more clearly in the text. Moreover, I would appreciate if the authors could state some specifics regarding their model in the main text. For example, they state that they estimated a linear mixed model with fixed effects on a university-level and random effects of the correlation (meaning a model with random intercept and random slope). As there are many different types of multilevel models, it would be nice to know immediately what type of model was chosen. The explanation and formula may remain in the supplementary material, but the main message should be delivered in the main text.

20. The authors should review their figure captions. I think that the caption should contain enough information to understand the figure without looking at the main text at the same time. Otherwise, the figure lacks necessary information and cognitive load is substantially increased. For example, the authors should add the information to figure 3 that the dotted black line represents the overall explanatory power of PEA in that the indirect effect (UKCAT \square PEA \square outcomes) is divided by the total effect (UKCAT \square outcomes). I made a similar observation with figures 4, and 5, where the explanation in the main text should be added to the figure caption. For example, "medical school performance (as a standardized z score)" (p. 17) or "the horizontal black dotted lines indicate the equivalent level of performance between those entrants from secondary schools at the lower decile of performance and those at the upper decile" (p. 17). This information is very useful in understanding the presented results.

21. Moreover, the authors should state their chosen deciles in the figure caption, even if these are "arbitrary points" (p. 40).

22. Finally, I think that the authors could explain their figures 4 and 5 a bit more in their main text. For example, they could add, for example, "as seen by the higher z scores of undergraduate medical school outcomes for lower levels of school level performance" to their sentence "...from schools with lower average attainment tended to have better subsequent scores in both knowledge and skills exams" (p. 17). I think that not all readers are that familiar with complex multilevel models, thus I would recommend describing the results in a way that is easy to comprehend.

Discussion

23. I was surprised that there were apparently no differences between mediation models for knowledge and skills (PEA mediated 43% in both cases). I would have expected different results for skills, and I think that possible differences were one of the main reasons for conducting separate models in the first place. However, the authors do not critically discuss these findings vis a vis the existing literature.

24. Again, the authors mention that they "were able to delineate the direct and indirect (mediational) effects of secondary school-level performance" (p. 20). However, as far as I know they did not investigate school-level performance as a mediator, but rather a direct predictor of medical school outcomes. Therefore, I would recommend rephrasing this section.

	25. I cannot fully accept the explanation of “positive influence of the university educational environment” (p. 20). The authors do not report the predictive value of PEA and UKCAT for medical school scores. Neither do they list average medical school outcomes across the years in their main text. Thus, they cannot conclude that PEA is not as important (only because it does not explain as much of the effect of UKCAT as it did before). Neither can they conclude that university environment has a positive influence, because we do not know whether achievements are “better” than in earlier years. It is also possible that students become more homogeneous over time (which implies reduced variance), without one particular group (e.g. of previously disadvantaged students) improving in terms of their medical knowledge and skills.
--	---

VERSION 1 – AUTHOR RESPONSE

Response to editorial comment:

Editorial comment:

Please revise your title to indicate the research question, study design, and setting. This is the preferred format of the journal.

Author’s response:

In line with the preferred format of the journal, the title has been revised to read “What is the effect of secondary (high) schooling on subsequent medical school performance? A national, UK-based, cohort study”

Comments: Reviewer #1

1) It would not be unusual for a skills-based final exam to be less reliable than a knowledge-based examination. Lower reliability has the effect of attenuating the magnitude of relationships and could explain the difference between skills- and knowledge-based assessments. At least, this must be acknowledged.

Author’s response:

This is a fair point, this is now acknowledged on page 23 of revised manuscript within the ‘strengths and limitations’ paragraphs of the Discussion section with appropriate references.

2) It would be helpful to get some sense of the differences among the medical schools in UKCAT and PEA. I would expect that some were more selective than others. Likewise, I suspect that the schools varied in the number of students they contributed to the analysis. How does this influence the results?

Author’s response:

Many medical schools use more than one method, at different stages in their selection process to widen participation and improve selection. Unfortunately, the specific methods used were not available in the data for this study and as such their differences cannot be studied from the data. For

example, the UKCAT may be used in selection as a 'borderline method' (to discriminate amongst a small number of applicants lying at a decision borderline, who are otherwise indistinguishable on the school's other selection criteria), 'factor method' (add an applicant's UKCAT score or a proxy for that score to the score the applicant obtains in the school's usual method of selection, to provide a total score), 'threshold method' (minimum or threshold UKCAT score adopted to create a hurdle that an applicant must cross to reach the next stage in the selection process) and 'rescue' (to compensate for an applicant's who would otherwise be rejected on account of their score on other selection criteria)[1].

With regard to PEA, Table 3 in the revised manuscript shows that over 90% of the entrants were selected with A-level grades of AAB and above. This means that A-level grades were homogenous across the different medical schools. With regard to the missing data, Table 1 in the revised manuscript shows the number of universities and students who participated in the study declined sharply in the fifth year. The knowledge based exams were less affected by this though compared to the skills based exams. Sensitivity analysis conducted to ascertain the impact of missingness in the results found no adverse effects. Please refer to page 24 of the revised manuscript and section 6 of the supplementary document for further details.

3) The outcome measures were standardized within school. This removes any variability associated with differences among them, although these differences are present in the predictor variables. How does this influence the results, particularly for those from less well performing secondary schools?

Author's response:

This is an interesting point. It should be noted that standardisation allows outcome scores from different medical schools to be compared on a common scale (mean zero and variance one). This in no way eliminates the variability (if any) across the different medical schools due to contextual factors. This is clearly demonstrated by the box plots of undergraduate year one knowledge-based exam in the supplementary document (Figure 1 with explanations on pages 2 and 3) which indicate that the relative distribution (i.e. relative size of box plots) between medical schools does not change much after standardisation. Thus the variability in the outcome measures were dealt with by use of multi-level models. For variability associated with predictor variables (especially related to less well performing schools), see response to comment (2) above and reply to comment (9) by reviewer (3).

4) The authors note this is not a typical cohort study in that students from the 2008 group enter and exit medical school at different times. It would be useful to provide more descriptive information (perhaps in the appendix) about which students (by PEA, UKCAT) enter sooner rather than later (do less able students enter later?).

Author's response:

This is unlikely to be an issue. The students entered medical school at the same time (2008) and followed up over the same duration of time (5 years), however the missing data for their knowledge and skills outcomes was a result of mainly medical schools deciding not to return data for that year rather than student exiting the study or dropping out as suggested by the term "attrition". The use of the term has now been dropped unless when explained in context as on pages 5 and 15. Also note that, based on description of missingness, the missing data mechanism is MCAR or potentially MAR.

The handling and effect of the missing data is mentioned in the discussion section on page 24 of the revised manuscript.

5) Why is there student attrition? Is attrition related to PEA or UKCAT scores?

Author's response:

Please refer to response to comment (4) above for information on whether attrition was related to PEA or UKCAT scores.

Comments: Reviewer #2

1) The high attrition rate is disappointing for year 5, and I wondered why the authors did not restrict the analysis to Years 1-3/4. I was unable to find an explanation for the high attrition, was it the unavailability of medical school assessments or did some of the 2008 cohort start medical school early? Please make this clearer.

Author's response:

The issue of attrition was also raised by reviewer # 1 in comment (2), (4) and (5) above. The cause is addressed under response to comment (4) by reviewer # 1. Indeed, the scale of missing data for the fifth year is concerning. This concern was dealt with by not excluding year 5 from analysis but by including it in the study and subsequently conducting a sensitivity analysis to determine whether this decision had an adverse impact on the results. The results of this sensitivity analysis included on page 24 of the revised manuscript and section 6 of the supplementary document indicate that the missing data did not have an adverse effect on the results of the study. Thus, we felt it was worthwhile to report the fifth year results.

2) Clinical skills assessments converted to a percentage score are invariably highly negatively skewed in medical education and converting to a zscore does not turn this into a normal distribution. Please provide some graphical illustration of the actual distributions of the scores you use.

Author's response:

This is an excellent point, the distribution of the knowledge-based outcome in year one of medical school training (as a motivating example for the other outcomes) is now presented in Figure 3 with explanation on page 5 of the supplementary document. Note also that medical schools were left to define what "skills" meant- it may be group tasks- generally anything other than a straightforward knowledge test but we are not clear about this. This is a potential limitation and is explained on page 23 of the revised manuscript.

3) The technical appendix is very lengthy and could benefit from an edit to make it more parsimonious. I expected more information about the differences in medical schools performance data and their distributions and less emphasis on statistical formulae. Is it possible that Figure 4 is incorrect? – it seems identical to Figure 3 and all other figures for "theory" (assume knowledge) and "skills" are different.

Author's response:

Whilst the supplementary document is unusually lengthy, we feel this is warranted by the nature of the methodology used in the study. To the best of our knowledge, this is the first study within the medical education that makes use of (multi-level) mediational analysis. For this reason, further detailed information regarding the methodology was provided in consideration of any reader who may want to follow closely how the methodology works. With regard to the distribution of the outcomes, more information is presented in Figure 3 with explanation on page 5 of the supplementary document in line with comment (2) above. Thank you for pointing out the error in Figure 4 (now updated to be Figure 6) in the supplementary document. We apologise for this inaccuracy. That has now been corrected.

4) The main body of the text could be more parsimonious as well. For example if you had no intention of using the sub-score of the UKCAT, why explain them in such detail?

Author's response:

Since BMJOpen is an international publication and most readers would be unfamiliar with UKCAT, we thought it best to give a detailed description of the components of the UKCAT. It is the scores from these components that were aggregated into the total UKCAT score used for the analysis. We are happy to take further advice on this and we do not object to shortening that section of the paper to make the paper more parsimonious.

5) I would suggest the following minor editorials changes prior to publication:

- Typo in the first sentence of the abstract if I am not mistaken: Should the third last word be at rather than an?

Author's response:

There is no typo in that sentence. The sentence reads correctly as far as we are aware.

- The first paragraph of "strengths of the study" needs as edit as well.

Author's response:

That section has been rewritten as advised.

- Page 13, line 50 the second this should probably be the

Author's response:

"The second aim of the study" is now changed to "The second aim of this study"

- Page 18, line 34 should read tend to

Author's response:

There is no typo in that sentence. The sentence reads correctly as far as we are aware.

- Be consistent in the use of knowledge and skills – theory and skills is used in some places and for some graph

Author's response:

All instances of "theory" have now been changed to "knowledge"-we apologise for this inconsistency which has now been corrected.

Comments: Reviewer #3

Abstract

1) I do not think that the abstract is entirely correct. Exploring "the mediating role played by the performance of a candidate's secondary school" (p. 2) is listed as a main objective.

However, the authors did not investigate school-level performance as a mediator, but rather a predictor of undergraduate outcomes (knowledge and skills). I would suggest rephrasing this section, for instance: "Moreover, we explore the impact of school-level performance on undergraduate outcomes (knowledge and skills) to inform selection policy".

Author's response:

We agree with this suggestion. That sentence in the abstract has now been revised as suggested.

In addition, the authors should state that they investigate knowledge and skills as two separate outcomes in their study. Currently, this distinction appears in the Results for the first time.

Author's response:

In the objectives section of the abstract, the sentence "subsequent undergraduate outcomes" has now been revised to read "subsequent undergraduate knowledge and skills-related outcomes analysed separately"

2) Under Methods, the authors could state that they used data from the 2008 medical school entry cohort for all purposes. I think that it is confusing to explicitly link "medical school outcomes" (p. 2), but neither UKCAT scores nor PEA to the year 2008. Moreover, I do not think that there is a sufficient distinction between "secondary school exam grades" and "school-level performance data" (p. 2). I was puzzled, since I expected a mediation including medical school outcomes, UKCAT, and PEA, but not necessarily school-level performance data. In accordance with "Objectives", I would recommend rephrasing the section, for example, "UKCAT scores and Prior Educational Attainment (PEA) were available for 2,107 students and were linked to medical school outcomes for each of the five years of medical school. PEA and school-level performance were based on school exam grades." Thus, it becomes clear that both, PEA and school-level performance are based on grades. It takes some time and space to explain the difference in the operationalization of these two, but I think that the main text is the place to do so rather than the abstract.

Author's response:

The method section of the abstract has now been amended to reflect accurately the modelling that was conducted within the word limit of the abstract as specified.

3) Similarly, I would state something like “knowledge-based undergraduate performance” rather than “exam performance” to maintain clarity in that “school exam grades” are not the same as (undergraduate) exams.

Author’s response:

The abstract has been amended to communicate this more clearly.

4) Moreover, the authors do not mention the mediation of UKCAT scores by PEA for skills-based exam performance. This should be mentioned, because it refers to the main research question.

Author’s response:

The abstract has been edited to reflect this suggested change. Note that, following corrections suggested post peer review, both knowledge and skills-based exams were found to be mediated by PEA. Therefore we now refer to these two outcomes as undergraduate academic performance in that section of the abstract to limit ourselves to the stipulated word limit.

Introduction

5) I am not sure why high grades are supposed to be connected to socio-economic status or even the type of school. Theoretically, students at underperforming schools, i.e. schools with a school-level performance below average, can achieve high grades and attend medical school. It seems clear to me though, why 80% of medical students stem from

20% of the country’s schools, if these schools belong to the high performance schools in the country, and most of their students achieve high grades. In my opinion, a major flaw in a lot of these estimations and policies is the misrepresentation of school-level performance as a sociocultural factor, which is one of the strengths of this study. Because a discounting procedure for individual applicants that is based on school-level performance, does not accurately consider the differences between schools and clustering effects of schools. Thus, I would suggest building a stronger argument for the differentiation of individuals and schools in “widening access” to medical education.

Because apart from mentioning in the introduction, the authors do not assess socio-economic differences and consequently, do not include it into their model.

Author’s response:

The premise of our argument which we did not convey fully in that section is that the 20% of the secondary schools we refer to are selective. These type of schools are better resourced compared to the non-selective schools. They are also highly attended by students in higher social economic backgrounds. Therefore differences in performance between selective and non-selective schools reflect, to a high degree, differences in material deprivation rather than intellectual ability of the students from those schools. This is attested to by the report we cited and the results from the UKCAT-12 study [2]. We have now edited that section to convey the message more clearly and cite

the UKCAT-12 study in that section as well. With regard to the clustering effects of the school, this was accounted for in all the models. All the models fitted were multi-level. In some instances, the multi-level models were changed to single level models where statistical testing provided no evidence for the effect of the cluster. The details on this may be seen in section 3.3 of the supplementary document. Moreover, in the discussion section we now cite the recent paper published by Kumwenda et al. in the BMJOpen which reported that state educated medical students outperformed privately educated ones, on average. This supports our argument in relation to socioeconomic advantage.

6) Another point refers to the similarities of PEA and UKCAT. I am not sure if the authors wanted to stress a difference by stating that “PEA has been demonstrated to have predictive validity for undergraduate medical school outcomes” (p. 4) and “Aptitude tests [...] have predictive validity for medical school outcomes” (p. 5). Maybe I have missed the point here, but as I see it, the authors point to potential similarities between both tests without additional value. To make this point more convincing, the authors could reference national data to compare entrants or applicants between schools with and without aptitude tests. If aptitude tests were invented to address the gap between socioeconomically advantaged and disadvantaged students or schools, then the gap between entrants for low-performance schools and high-performance schools should shrink after aptitude tests have been introduced. Unless there are other factors, such as sociocultural influences that are more important when deciding to apply for medical school.

Author’s response:

This is an interesting point that underscores the complexity of selection. Medical schools differ in the way they use the UKCAT for selection (see response to comment (2) by reviewer #1). In addition to UKCAT, medical schools still use PEA to some degree along other selection criteria like interviews and references. Some medical schools even use a different aptitude test, the BMAT. For these reasons, it is a huge challenge to obtain national data that would enable comparison between selection done purely based on aptitude tests and selection done purely based on PEA alone. However, previous work suggested that under-represented groups were less disadvantaged when applying to medical schools using the UKCAT in a relatively robust way [3] and that the UKCAT may be relatively less sensitive to the school -type attended, compared to traditional academic attainment [4]. However subsequent longitudinal analysis suggested that such effects did not have a consistent impact on reducing disadvantage for certain underrepresented groups [5]. Aptitude test scores also tend to correlate to a moderate degree, with educational attainment. Certainly, we agree with the reviewer that it would be interesting to explore whether these differences were less marked within medical schools that had a more robust use of aptitude tests, but this is beyond the scope of the present study.

Most studies, following the introduction of the UKCAT, have endeavoured to assess the performance of either PEA or UKCAT while controlling for the effects of the other. In this study, our intention was not to take the discussion further by not just obtaining the effect of the UKCAT controlled for the effect of PEA but to determine what proportion of the UKCAT may be explained by PEA since they are both used in selection.

Additionally, the lack of content validity could be discussed separately. A first step could be to provide measures of correlation between both, UKCAT and PEA or an estimate of their predictive value regarding undergraduate achievement.

Author’s response:

Content validity (the extent to which UKCAT or PEA measures what it is intended to measure) was not within the scope of this study. The scope of the study was the predictive validity of UKCAT and

PEA (predictive power for future undergraduate knowledge and skills based exams). As a preliminary step towards this, we now provide as suggested, correlation coefficients between UKCAT/PEA and future knowledge and skills based undergraduate exams in Table 3 of the supplementary document.

7) Unfortunately, I am not very familiar with the educational system in the UK. Therefore, I would appreciate if the authors simply stated the order of these exams and tests at one point. They could simply outline that UKCAT are taken before final exams (secondary school exams) and before medical school, i.e. in the summer of the year before entry, I believe. In addition, they could make clear that they estimated PEA on standard exams like GCSE and A-levels in contrast to general school performance (predicted grades). In their present analysis, if I am correct, the authors investigate PEA, based on final secondary-school exams, as a mediator of UKCAT scores, which were assessed at an earlier time, on medical school performance (undergraduate performance at the end of the year). I think that it would be much easier for an international audience to comprehend the mediation models and the interplay of these variables, if the authors provided a clear description and a flow.

Author's response:

Your assessment of the ordering of the UKCAT, PEA and undergraduate medical school is correct. This information was initially put in section 2.1 of the supplementary document. Based on your observation, this has now been moved to page 7 of the revised manuscript.

8) I have a small issue with the wording in the main text. The authors discern school-level performance (ordinal grades like AAA, ABB) and PEA (latent variable) as indicators of school performance on a school-level and an individual level. However, this could be solved by introducing a fixed set of words from the beginning. If they referred to school-level performance and individual performance or PEA, for example, it would be much easier not to confuse these two levels when reading the methods, and results. Similarly, the authors could decide to use "medical school" or "undergraduate" outcomes, such as "undergraduate knowledge" and "undergraduate skills". However, I think that they should not switch between these phrases within the text. In my eyes, the sophisticated analysis is much easier to comprehend if presented consistently.

Author's response:

The different terms are now clearly defined as suggested within the text and used consistently. We use the terms "undergraduate knowledge and skills-based outcomes (assessment / exams)" or undergraduate medical school outcomes (when not being specific). This has been done in order to differentiate it from (secondary) school level performance and PEA. Hopefully, this will now help with improving the readability of the paper.

9) I do not see why the influence of school-level performance on undergraduate's achievement should necessarily inform policy on grade discounting. In the current sample, the authors investigate students from poorly performing schools that have already taken UKCAT and "survived" medical school for at least a year. This could mean that they have a different outlook on medical school

compared to their peers or that they feel the need to prove themselves among socio-economically advantaged peers, which would explain their high scores and test results. Thus, it is hard to compare them to the average at their secondary schools. If these students already represent “the best” at their school, it could be possible that grade discounting for their secondary school does not lead to the expected consequences. Instead, I think that it could be interesting to compare UKCAT scores of students from schools with different performance levels and estimate the predictive value of these scores for academic success. For analytical purposes, the authors could suggest a comparison of schools, e.g. between two or three groups (+1 SD and -1SD of school-level performance), and report coefficients for these groups. Thus, they could also report the number of students at AAA and AAB levels in each group, and so on to further elucidate these differences.

Author’s response:

This is an interesting observation. It is worth noting that grouping schools based on +1SD and -1SD may not reflect the true similarity between schools. This is because standardisation offers change of scale and interpretation based on a common metric. Please see response to comment (3) by reviewer #1. We propose the use of MMREM (Multiple Membership Random Effect Model) described in response to comment (16) below. The challenges associated with the proposed approach are also discussed in the response. Nevertheless, it is possible to crudely investigate whether UKCAT scores may vary based on secondary school level performance. This has been done by conducting a one factor Anova multiple comparison analysis by grouping schools in three (categories) based on their average secondary school performance. The details and results for the test are included (in Figures 2 and Table 2) on pages 4 to 5 of the supplementary document. Further, the impact of the secondary school categorisation on future medical school performance was investigated using a multi-level model. The results are available in Tables 11 and in the supplementary document.

10) I would like to know how many UKCAT-consortium medical schools there are. It should be possible to report how many schools and students are represented by the 18 schools and 2,107 students included.

Author’s response:

In 2007 UKCAT testing cycle from which data was derived, there were 26 UKCAT-consortium medical schools. Therefore this study comprised of 69% of the 26 UKCAT-consortium medical school as at that time. The study included all applicants who sat for the UKCAT and selected to join one of the 18 UKCAT-consortium undergraduate medical schools. This is now included under the data section of the revised manuscript on page

Methods

11) Although the authors did not recruit the sample themselves, they should provide some information regarding data privacy standards, and ethical considerations. The UKCAT consortium may have addressed these points in the past, but it is important to read and comprehend the researchers’ point of view and possible concerns.

Author’s response:

The data used for this study was anonymised. From the data, no piece of information can be used to identify a secondary school, medical school or individual. As such, the identity of the participants is fully protected. All participants consented to the collection of the data for research. This information is now explicitly made available in the declaration section of the revised manuscript.

12) The authors could state the exact value for the “relatively low attrition rate” (p. 8) in the text.

Author’s response:

Please see response to comment (4) by reviewer # 1.

13) I am curious: Was it possible to assess sociodemographic information like race/ethnicity, gender or age? If not, this should be discussed as a limitation.

Author’s response:

This study is a follow up to a previous study conducted by the authors [6] which assessed the effect of the UKCAT controlled for PEA and other sociodemographic variables (among them sex, age, socioeconomic status and age). In the study, the effects of the sociodemographic variables did not impact on the results and conclusions made. Moreover, currently the sociodemographic variables of age gender and ethnicity are not used in the UK to based selection decisions on. This is why they were not included in the models for this particular study as there were no immediate implications for selection policy.

14) One of my main concerns refers to the centering process used in this study. I appreciate that the authors provide information on the preparation of their data set, but I am not sure I understand every step. For example, medical school outcomes were assessed as z-scores for each school (group-mean centering), so that each students z score can be compared to its peers, but not across schools. Thus, differences in “high performance” medical schools or cohorts at some schools may lead to similar z scores compared to other schools with lower grades due to the reduced variance of the subsample. Unfortunately, the authors do not provide variance estimates for each medical school so it remains unclear. However, they state intraclass correlations (ICC) of outcomes in their appendix. First, I think that this information is important and belongs in the Methods’ main text. Second, I am not sure if the ICC are based on percentage scores or z scores.

Author’s response:

The raw mean and variance estimates for each school for the knowledge-based exam outcome in year one (taken as a motivating example for other outcomes) is now included (in Figure 3) in the supplementary document. Also, the ICCs in Table 7 of supplementary document are based on z-scores rather than percentage scores. Even if percentage scores were used, the ICCs obtained would not differ significantly as standardisation only results in change of scale and not change in association of the variables under consideration. Please refer to response to comment (3) by reviewer #1. Our motivation for including the ICC in supplementary document was to make the main manuscript easier to read for the less statistically erudite reader. We are still convinced that this approach is best considering the broad readership of the BMJ Open. We are however to happy to take further advice on this and make changes accordingly.

15) In contrast, the UKCAT scores were standardized “for all candidates at the year of sitting” (p. 10) as were PEA factor scores “for all applicants” (p. 11) (grand-mean centering). However, this means that UKCAT and PEA are compared between all of the students in the sample regardless of their secondary school and their medical school. Thus, it is not possible to deduce whether a student at a poorly performing school belonged to the best in his or her secondary school (which is problematic regarding the second research question), and whether there are differences in PEA and UKCAT scores between medical schools that could also correspond with school-level performance. To investigate these differences, the authors could also provide ICC estimates for raw scores. Finally, school-level performance is stated as a school-level grade average without centering. Thus, differences in school-level performances reflect mean differences for a student, irrespective of the variance of performance at each institution. Thus, school-level performance cannot be easily compared to PEA, UKCAT scores or outcomes (all of which are centered in one way or another). I wonder if the results differed if grand-mean centering was applied to school-level performance, so that the mean value of 225.18 points referred to “0” or average performance, and higher and lower scores represented “higher” and “lower” performance across schools.

Author’s response:

This is an acute observation which highlights the complexity of the data. It is indeed the case that care is taken to account for variation in outcomes at the level of the medical school. This is done by standardising the undergraduate outcomes at the level of the medical school and utilising the medical school as a higher level variable in a multi-level model in addressing the first and second objective of the study. This enables the ICCs to be computed if desired. However as noted by the reviewer, it is possible that UKCAT and PEA may have differences associated with secondary school level performance prior to enrolment at a medical school. The modelling of these differences would need a Multiple Membership Random Effect Model (MMREM) in order to obtain the ICC as suggested. The use of MMREM would be necessitated by the fact that it is possible for the lower level unit, secondary school, to belong to more than one medical school. This is clearly seen by imagining two students from the same secondary school being enrolled in two different medical schools. The main hurdle in this approach is that some secondary schools are represented by a count of less than 2 in each medical school. This makes it impossible to estimate variability at this level (due to the small sample size) so as to compute ICC as suggested. Therefore we acknowledge this is a limitation of the study.

Results

16) This points to the multilevel models, in general. In their technical appendix, the authors explain that neither effect varied significantly between medical schools. However, they do not report their findings for educational achievement, school-level performance, and so on in the main text. While I appreciate the detailed explanation in the supplementary material, I would recommend including more information in the main text to guide the reader through each decisional step.

Author’s response:

Our motivation for including the more technically advanced aspects of the study in supplementary document was to make the main document easier to read for the less statistically erudite reader. We are still convinced that this approach is best considering the broad readership of the BMJOpen. We are however happy to take further advice on this and make changes accordingly

17) I am confused by the choice of words. First, there are “no statistically significant clustering effects by university” (p. 13), but then there is “variation in outcomes between universities” (p. 13) that needs to be accounted for. I assumed that in both cases, universities represented the higher level, and medical school outcomes represented individual “undergraduate achievement” (p.13). If the authors refer to schools instead of universities or to school-performance instead of undergraduate achievement, they should say so more clearly in the text. Moreover, I would appreciate if the authors could state some specifics regarding their model in the main text. For example, they state that they estimated a linear mixed model with fixed effects on a university-level and random effects of the correlation (meaning a model with random intercept and random slope). As there are many different types of multilevel models, it would be nice to know immediately what type of model was chosen. The explanation and formula may remain in the supplementary material, but the main message should be delivered in the main text.

Author’s response

There were two types of models fitted to address the objectives listed on page 8 of the revised manuscript. The first objective was addressed by first fitting a multi-level mediation model with UKCAT, PEA and undergraduate knowledge/skills based outcomes. The statistical tests relating to these models in section 3.3 of the supplementary document demonstrated that there was no clustering effect of the university hence the statement in the revised manuscript on page 13 (to 14). This is explained on that page. It is also communicated on that page that extra details on the statistical tests employed are available in the relevant section in the supplementary document.

With regard to the second objective, a series of linear mixed models with a random slope for each university were fitted. This information is available on page 14 of the revised manuscript and section 4 of the supplementary document. This is also clearly communicated in the main manuscript.

18) The authors should review their figure captions. I think that the caption should contain enough information to understand the figure without looking at the main text at the same time. Otherwise, the figure lacks necessary information and cognitive load is substantially increased. For example, the authors should add the information to figure 3 that the dotted black line represents the overall explanatory power of PEA in that the indirect effect (UKCAT PEA outcomes) is divided by the total effect (UKCAT outcomes). I made a similar observation with figures 4, and 5, where the explanation in the main text should be added to the figure caption. For example, “medical school performance (as a standardized z score)” (p. 17) or “the horizontal black dotted lines indicate the equivalent level of performance between those entrants from secondary schools at the lower decile of performance and those at the upper decile” (p. 17). This information is very useful in understanding the presented results.

Author’s response:

These suggested changes have been adopted

19) Moreover, the authors should state their chosen deciles in the figure caption, even if these are “arbitrary points” (p. 40).

Author's response:

These suggested changes have been adopted

20) Finally, I think that the authors could explain their figures 4 and 5 a bit more in their main text. For example, they could add, for example, "as seen by the higher z scores of undergraduate medical school outcomes for lower levels of school level performance" to their sentence "...from schools with lower average attainment tended to have better subsequent scores in both knowledge and skills exams" (p. 17). I think that not all readers are that familiar with complex multilevel models, thus I would recommend describing the results in a way that is easy to comprehend.

Author's response:

These suggested changes have been adopted

Discussion

21) I was surprised that there were apparently no differences between mediation models for knowledge and skills (PEA mediated 43% in both cases). I would have expected different results for skills, and I think that possible differences were one of the main reasons for conducting separate models in the first place. However, the authors do not critically discuss these findings vis a vis the existing literature.

Author's response:

The proportion of UKCAT explained by PEA differs for the two undergraduate medical school outcomes as may be seen in the Figure below. The proportions explained were much larger for undergraduate knowledge than skills-based outcome. This is not clearly seen in Figure 3 of revised manuscript because the proportions are not shown on the same axis. We thought it best to separate them as we were interested in showing the uncertainties (confidence intervals) associated with the proportions. As we may be seen from Figure 3 in the revised manuscript, higher level of uncertainty are associated with undergraduate skills-based outcomes. If the proportions for the two outcomes with their associated outcomes had been shown on the same axis, the graph would have been cluttered. The value of 43% (dotted horizontal line) was arbitrarily chosen to show difference in proportion between the first two years and the rest of the years of medical school. To confirm that the proportion of predictive power of the UKCAT, explained by PEA, for the two undergraduate outcomes were indeed statistically significantly different. A test of proportions was conducted whose results are described on page 19 (and included in Table 10) of the supplementary document.

22) Again, the authors mention that they "were able to delineate the direct and indirect (mediational) effects of secondary school-level performance" (p. 20). However, as far as I know they did not investigate school-level performance as a mediator, but rather a direct predictor of medical school outcomes. Therefore, I would recommend rephrasing this section.

Author's response:

This has been edited to reflect what was done more accurately as suggested.

23) I cannot fully accept the explanation of “positive influence of the university educational environment” (p. 20). The authors do not report the predictive value of PEA and UKCAT for medical school scores. Neither do they list average medical school outcomes across the years in their main text. Thus, they cannot conclude that PEA is not as important (only because it does not explain as much of the effect of UKCAT as it did before). Neither can they conclude that university environment has a positive influence, because we do not know whether achievements are “better” than in earlier years. It is also possible that students become more homogeneous over time (which implies reduced variance), without one particular group (e.g. of previously disadvantaged students) improving in terms of their medical knowledge and skills.

Author’s response:

The statement in the discussion is not firm in its conclusion rather we offer possible plausible explanations as to why the gap in performance between students from high and low performing secondary schools seems to narrow over time in medical school. This is subject to further research, that section has now been edited to convey that more clearly. As suggested, we now acknowledge in the revised manuscript that it is also possible that the undergraduate medical school performance narrows as a result of students becoming more homogenous over time. It is important to note though, since the content of the undergraduate medical school exams differ from year to year, it would be ill advised to analyse their means differences between the years as suggested.

References

1. Adam J, Dowell J, Greatrix R. Use of UKCAT scores in student selection by UK medical schools, 2006-2010. *BMC medical education*. 2011 Nov 24;11(1):98.
2. McManus IC, Dewberry C, Nicholson S, Dowell JS. The UKCAT-12 study: educational attainment, aptitude test performance, demographic and socio-economic contextual factors as predictors of first year outcome in a cross-sectional collaborative study of 12 UK medical schools. *BMC medicine*. 2013 Nov 14;11(1):244.
3. Tiffin PA, Dowell JS, McLachlan JC. Widening access to UK medical education for under-represented socioeconomic groups: modelling the impact of the UKCAT in the 2009 cohort. *BMJ*. 2012 Apr 17;344:e1805.
4. Tiffin PA, McLachlan JC, Webster L, Nicholson S. Comparison of the sensitivity of the UKCAT and A Levels to sociodemographic characteristics: a national study. *BMC medical education*. 2014 Jan 8;14(1):7.
5. Mathers J, Sitch A, Parry J. Population-based longitudinal analyses of offer likelihood in UK medical schools: 1996–2012. *Medical education*. 2016 Jun 1;50(6):612-23.
6. Tiffin PA, Mwandigha LM, Paton LW, Hesselgreaves H, McLachlan JC, Finn GM, Kasim AS. Predictive validity of the UKCAT for medical school undergraduate performance: a national prospective cohort study. *BMC medicine*. 2016 Dec;14(1):140.

VERSION 2 – REVIEW

REVIEWER	Samuel Tomczyk University of Greifswald, Department of Health and Prevention, Greifswald, Germany
REVIEW RETURNED	27-Feb-2018

GENERAL COMMENTS	In their revision, the authors have addressed the most important concerns raised by the reviewers. I think that the changes have improved the manuscript, e. g. consistent wording (see “knowledge” and “skills”) and additional information on the selection process of the investigated medical schools in the UK make it much more comprehensible, particularly for an international audience. Also, the discussion benefits from the inclusion of alternative explanations, and possible limitations regarding medical school outcomes. I appreciate the profound response to the reviewers, and while I retain some questions regarding the adequacy of the methodological approach, I think that the extensive supplementary material offers a well-structured and accessible documentation of the process and sufficient explanations for the current decision. Given that the supplementary material will be available online, I do not think that its length is problematic. Minor suggestions:  1. ... tend to achieve better undergraduate exam results (p. 19) instead of ... “tend achieve better undergraduate exam results” 2. Please include the explanation of the dotted black line in the figure caption of figure 3. The captions of figures 4 and 5 explain the dotted black lines. Figure 3, however, does not. It seems arbitrary, as it is comparable to the proportion of skill-based exams in the first year, but no other coefficient. If the threshold of 43% was chosen because of this similarity, this could and should be stated (in the figure caption, at least).
---

REVIEWER	Associate Professor Deborah O'Mara Sydney Medical School University of Sydney Australia
REVIEW RETURNED	03-Mar-2018

GENERAL COMMENTS	Congratulations on an outstanding and thorough piece of research on selection for medical education. Your manuscript has been subject to very detailed scrutiny and you have made changes that will significantly improve the accessibility of the paper. Thank you for addressing the reviewers' concerns in such detail. Personally I would have excluded Year 5 due to the missing data and/or used strategies to ensure such large missing data did not occur. However, you do provide a justification and I accept your argument.
---

VERSION 2 – AUTHOR RESPONSE

Response to editorial comment:
Editorial Request:

The methods section of the abstract is very short. Can this be expanded to provide a more detailed summary of what you did? We recommend, but do not insist, that you use the abstract sub-headings recommended in our instructions for authors for research articles.

Author's response:

Thank you for your recommendation. The method section of the abstract has now been expanded to provide more detail concerning the analyses done in the study. This has been done whilst keeping the word count to the required maximum of 300.

Comments: Reviewer #2

Congratulations on an outstanding and thorough piece of research on selection for medical education. Your manuscript has been subject to very detailed scrutiny and you have made changes that will significantly improve the accessibility of the paper. Thank you for addressing the reviewers' concerns in such detail. Personally I would have excluded Year 5 due to the missing data and/or used strategies to ensure such large missing data did not occur. However, you do provide a justification and I accept your argument.

Author's response:

Thank you for taking the time to read through the manuscript and for your remarks.

Comments: Reviewer #3

In their revision, the authors have addressed the most important concerns raised by the reviewers. I think that the changes have improved the manuscript, e. g. consistent wording (see "knowledge" and "skills") and additional information on the selection process of the investigated medical schools in the UK make it much more comprehensible, particularly for an international audience. Also, the discussion benefits from the inclusion of alternative explanations, and possible limitations regarding medical school outcomes. I appreciate the profound response to the reviewers, and while I retain some questions regarding the adequacy of the methodological approach, I think that the extensive supplementary material offers a well-structured and accessible documentation of the process and sufficient explanations for the current decision. Given that the supplementary material will be available online, I do not think that its length is problematic.

Minor suggestions:

1. ... tend to achieve better undergraduate exam results (p. 19) instead of ... "tend achieve better undergraduate exam results"

Author's response:

Thank for highlighting this error. This has now been corrected.

2. Please include the explanation of the dotted black line in the figure caption of figure 3. The captions of figures 4 and 5 explain the dotted black lines. Figure 3, however, does not. It seems arbitrary, as it is comparable to the proportion of skill-based exams in the first year, but no other coefficient. If the threshold of 43% was chosen because of this similarity, this could and should be stated (in the figure caption, at least).

Author's response:

Thank you for your observation. The caption for Figure 3 has now been edited to reflect the suggestion.